# Personal and relational changes following deep brain stimulation for treatment-resistant depression: A prospective qualitative study with patients and caregivers

Cassandra J. Thomson[1,2]*, Rebecca A. Segrave[1], Paul B. Fitzgerald[3], Karyn E. Richardson[1], Eric Racine[4,5,6], Adrian Carter[1]

1 School of Psychological Sciences, Turner Institute for Brain and Mental Health, Monash University, Clayton, Victoria, Australia, 2 Wicking Dementia Research and Education Centre, University of Tasmania, Hobart, Australia, 3 School of Medicine and Psychology, Australian National University, Canberra, Australia, 4 Pragmatic Health Ethics Research Unit, Institut de recherches cliniques de Montréal, Montreal, Canada, 5 Departments of Medicine & Social and Preventive Medicine, Université de Montréal, Montreal, Canada, 6 Departments of Neurology and Neurosurgery & Medicine, and Biomedical Ethics Unit, McGill University, Montreal, Canada

* cassandra.thomson@utas.edu.au

**Data Availability Statement:** Given the personal nature of the information included within the

## Abstract

### Objective

Deep brain stimulation (DBS) and whether it alters patient personality is a much-debated topic within academic literature, yet rarely explored with those directly involved. This study qualitatively examined how DBS for treatment-resistant depression impacts patient personality, self-concept, and relationships from the perspectives of both patients and caregivers.

### Methods

A prospective qualitative design was used. Eleven participants were included (six patients, five caregivers). Patients were enrolled in a clinical trial of DBS of the bed nucleus of the stria terminalis. Semi-structured interviews were conducted with participants before DBS-implantation and 9-months after stimulation-initiation. The 21 interviews were thematically analysed.

### Results

Three primary themes were identified: (a) *impact of mental illness and treatment on self-concept;* (b) *device acceptability and usability*, and (c) *relationships and connection*. Severe refractory depression had profoundly impacted who patients were, how they viewed themselves, and the quality and functioning of their relationships. Patients who benefited from DBS felt reconnected with their premorbid self, yet still far from their ideal self. While reductions in depression were broadly beneficial for relationships, the process of adjusting relationship dynamics created new challenges. All patients reported recharging difficulties and challenges adapting to the device.

interview transcripts and in order to maintain participant confidentiality, full data is not freely available. All relevant data are within the paper and its Supporting Information files. Any enquiries or requests for data access can be directed to the Monash University Human Ethics Research Committee: muhrec@monash.edu.

**Funding:** This work was supported by an NHMRC grant (1077859). AC is supported by an Australian National Health and Medical Research Council Career Development Fellowship (1123311). PBF is supported by a NHMRC Leadership award (1193596). RAS is supported by the David W. Turner Endowment Fund. ER is supported by a Fonds de recherche du Québec—Santé career award. CJT received an Australian Postgraduate Award scholarship to support her during her doctoral studies. The funders had no role in study design, data collection and analysis, decision to publish, or preparation of the manuscript.

**Competing interests:** I have read the journal's policy and the authors of this manuscript have the following competing interests: PBF has received equipment for research from MagVenture A/S, Nexstim, Neuronetics and Brainsway Ltd and funding for research from Neuronetics. He is a founder and board member of TMS Clinics Australia and Resonance Therapeutics. This does not alter our adherence to PLOS ONE policies on sharing data and materials. None of these commercial relationships were related to the current study or influenced the study design, data collection or analysis, decision to publish, or preparation of the manuscript. There are no patents, products in development or marketed products associated with this research to declare.

## Conclusions

Therapeutic response to DBS is a gradual and complex process that involves an evolving self-concept, adjusting relationship dynamics, and growing connection between body and device. This is the first study to provide in-depth insight into the lived experience of DBS for treatment-resistant depression. Patient and caregiver narrative accounts should be routinely collected to guide more person-centred DBS clinical interventions.

## Introduction

Deep brain stimulation (DBS) is a neurosurgical procedure being trialled in individuals with treatment-resistant depression (TRD). The procedure involves implanting electrodes in specific brain regions thought to be associated with depression psychopathology. Continuous electrical pulses are sent from a battery (implantable pulse generator; IPG) located in the patient's chest to the brain via subcutaneous leads. A range of target regions have been trialled for TRD, including the subcallosal cingulate gyrus [1, 2], ventral capsule/ventral striatum [3, 4], medial forebrain bundle [5, 6] and nucleus accumbens [7, 8]. The bed nucleus of the stria terminalis (BNST) has been a recent target of interest [9, 10] and is the implantation site for the present sample. DBS has demonstrated capacity to significantly and effectively alleviate depressive symptoms [11]. However, response and remission rates vary considerably across studies and optimal patient characteristics, stimulation parameters, and implantation sites remain under investigation [12].

While the efficacy and safety of DBS for TRD continues to be investigated via clinical trials, with mixed results [13], there have been no in-depth studies investigating patients' experience of DBS and their perspectives on the psychosocial changes they undergo. The impact of DBS on caregivers and family has also been under-investigated, from both a quantitative and qualitative perspective. Qualitative studies involving DBS candidates with TRD have been limited to questions of decision-making, capacity to consent [14–16], and attitudes towards emerging closed-loop systems [17]. These are important ethical issues to consider, particularly when offering an experimental treatment to potentially vulnerable individuals [18, 19]. But they do not provide insight into the lived experience of DBS or its broader psychosocial implications.

Qualitative studies with patients who have undergone DBS for other clinical indications, such as Parkinson's disease (PD), have revealed important insights using this methodological approach [20]. In PD, patients and caregivers describe both positive and negative changes in patient personality emerging after DBS (e.g., more fun, open, talkative; more aggressive, selfish, quiet) [21]. With improved PD symptoms, patients and caregivers can feel the patient's *old self* has been restored [22]. However, when patients experience unintended side-effects (e.g., irritability, compulsive behaviours) spouses can feel they are no longer married to the same person [23]. Some patients report difficulty accepting the implanted electrical device psychologically and experience altered body image, while caregivers can feel "lost" when their partner no longer depends on them [24]. In obsessive-compulsive disorder (OCD), patients describe post-DBS changes as being more or less aligned with their perceived true self and needing to *get used to* how they are now or needing to *find out* who they are without OCD [25].

These complex and highly nuanced psychosocial experiences are not captured by quantitative psychopathology and assessment tools used in clinical trials. However, they can have substantial implications for recovery and patient and caregiver wellbeing. Therefore, the aim of this study was to qualitatively examine how DBS for TRD impacts patient personality, self-

concept, and relationships from the perspectives of both patients and caregivers as they prepare and adjust to life with DBS.

## Materials and methods

This exploratory study used a prospective qualitative design. Methods are reported according to COREQ guidelines for qualitative research [26]. Ethics approval was obtained from Monash University Human Research Ethics Committee (CF16/1888-2016000963) and all participants provided written informed consent. Separate data findings derived from these interviews on the topics of informed consent, participant expectations, and subjective patient outcomes are reported elsewhere [27].

### Participants

Consecutive sampling was used to recruit individuals actively enrolled in a clinical trial of DBS for TRD who were awaiting surgery (Australian New Zealand Clinical Trials Registry: ACTRN12613000412730) (Clinical trial details, including inclusions/exclusion criteria, participant characteristics, surgical information, and full psychometric outcomes, will be presented in a forthcoming publication. Any correspondence regarding the clinical trial should be directed to paul.fitzgerald@anu.edu.au.). Participants were recruited by the first-author after permission to be contacted was granted via the clinical trial co-ordinator. Verbal and written study information was provided to the candidates and their respective caregivers. All who were approached chose to participate, including: six DBS candidates (5 women, 1 man, $M_{age}$ = 52 years, range = 26–73 years) and five caregivers (4 spouses, 1 parent, 2 women, 3 men, $M_{age}$ = 59.5 years, range = 45–75 years). One candidate lived alone and chose to participate independently. One DBS candidate who completed a preoperative interview did not complete a postoperative interview as it was deemed too burdensome for the patient. The current sample represents the majority, but not complete clinical trial sample, as the current study commenced after the trial began. DBS candidates had a long history of TRD (time since diagnosis: $M$ = 18 years, range = 8–42 years) and met Stage V criteria of the Thase and Rush [28] classification for TRD. None of the DBS candidates were engaged in paid employment. The DBS candidates and caregivers had long-term relationships, spanning decades ($M$ = 37 years, range 24–50 years).

### Procedure

One-on-one, semi-structured interviews were conducted between 2016 and 2019 with the first-author, a female psychologist with training in qualitative methods and experience interviewing DBS patients, families, and clinicians. Interviews were conducted face-to-face (at participants' home, research centre) or via telephone or video-conference for participants living interstate. Interviews were conducted separately with patients and caregivers to allow open discussion. Pre-surgery interviews occurred 3 to 15-weeks prior to surgery ($M$ = 46 minutes, range = 34–58 minutes) and explored participants' knowledge of DBS and beliefs about how the patient's personality, self, and relationships may be affected (see S1 File for interview schedules). Participants were not provided with a definition or description of 'personality'. We were interested specifically in their personal experiences of personality and identity change. This was done to avoid restricting participants' responses to a particular conceptual framework. Participants then underwent DBS surgery and were implanted with Medtronic Activa 3389 electrodes in the BNST. After a recovery period, a pseudorandomised schedule of active or sham (control condition) stimulation commenced, with participants blinded. Over five months, five stimulation settings were trialled: one inactive, two low-level (2 volts), and two

moderate-level (4 volts). Following this, stimulation continued in an open-label manner, with parameters optimised according to individual responses. Any existing treatments (medications/psychotherapy) were kept constant during the blinded phase with participants able to make changes during the open-label follow-up. Post-surgery interviews occurred 9 to 11-months after stimulation initiation ($M$ = 55 minutes, range = 36–86 minutes), roughly 3-months into the optimisation phase. These explored experiences living with DBS and the perceived personal and relational impact. Field notes were maintained during the interview process. Audio-recordings were transcribed verbatim by a professional transcription service, then reviewed for accuracy (by CT) and de-identified using pseudonyms.

### Data analysis

Interview transcripts were entered into NVivo 12 software (QSR International) and analysed using thematic analysis [29]. This is an iterative and inductive process that allows novel themes and patterns of meaning to develop from the data. Data was analysed according to the six-step process outlined by Braun and Clarke [29]. All interviews were coded by CT. Data saturation, the stage at which no novel themes were evident (commonly used to determine sufficient sample size in qualitative research), was reached at interview 9 of both the pre- and post-DBS interviews. Approaches to increase the quality and credibility of the results included cross-coding a subset (6) of interviews with the research team (AC, RS) to ensure consistency in coding. Any disagreements in coding were discussed until a consensus was reached. Triangulation, where a phenomenon is examined from different perspectives, was achieved by inclusion of a diverse research team (clinical psychology, neuropsychology, neuroethics, social science).

## Results

Thematic analysis revealed three primary themes relevant to the current paper. Primary themes developed longitudinally, with secondary themes reflective of specific time points (before/after surgery). Themes are illustrated using representative quotes in Tables 1–6. Both patient and caregiver perspectives are represented across all themes.

### Impact of mental illness and treatment on self-concept

Prior to DBS, participants expressed how severe and persistent depression had obscured defining aspects of the patient's self and personality (Table 1). These included traits such as being outgoing, confident, and a "bit of a joker". Three patients had previously enjoyed fulfilling careers in health care. These caring professions were integral to patients' identities and their inability to perform them due to mental illness was a significant loss. Patients' interest and ability to engage in meaningful and rewarding activities, such as intellectual and creative pursuits, were profoundly impaired and detrimental to self-worth.

Years of depression had created a sense of an absent person and life, with patients merely existing. This was pronounced in patients receiving regular ECT, which caregivers characterised as a means of keeping the patient alive rather than a treatment that remediated their depression. When asked if they thought DBS could change the patient's personality, most participants did not expect or were unconcerned that it would. Some hoped regained wellness would restore their premorbid personality. Others expressed concern DBS may alter personality in an undesirable way, but felt comfortable that adjusting or ceasing stimulation would resolve this, or otherwise they would simply "have to cope". These concerns were minor compared with participants' primary preoccupation: would the intervention be effective and how would they cope if it wasn't?

**Table 1. Impact of mental illness and treatment on self-concept–Pre-DBS.**

**Defining aspects of self and identity inhibited**

 Caregiver 2: *She was very independent and very free spirited and quite fiery as well [laughs], but that's all changed. She's just very passive, doesn't make decisions.*

 Caregiver 4: *She was confident, she was uplifting and energetic...but for quite some time now that's just not there.*

 Caregiver 5: *Yeah, she used to be fantastic in the work she did and she just lost all that confidence and it's just been a waste of her life for her and that's...been very hard to take.*

**Long-term depression creates an absence of person and life**

 Patient 3: *It's not just impacted on my ability to work; it's impacted on my ability to be a person.*

 Patient 5: *[I hope] to be able to live my life rather than existing because that's all I feel I'm doing now, just existing and not living it.*

 Caregiver 2: *[ECT] doesn't really lift her up...it's just existing.*

 Caregiver 3: *[With ECT] she ended up pretty much like a zombie...like the walking dead, it was horrible.*

**Potential impact of DBS on personality and self**

 Patient 1: *Depression itself kind of changes who you are so I suppose logically...that might also be a consequence but that's really to do with, that would be to do with resolving the depression, not with the surgery itself, the procedure itself.*

 Caregiver 2: *After such a long period of time it's hard to remember what that personality was. I think it certainly will change her personality compared to what it is now...well the hope is it actually restores what it was.*

 Patient 3: *No, I'm not concerned that it's going to change my personality. That's, again, one of the very common misnomers, that people say when you take antidepressants it alters who you are as a person or your personality traits...It might make you feel better or worse...but it's not changing who you are, that's a response to a symptom.*

 Patient 6: *I could come out the other side like Jekyll and Hyde...This thought occurred to me yesterday. What happens if...I come out the other side and I'm not the same person? I don't have the same values...How are we going to cope with that? We'll just have to cope...we'll just have to...work it out.*

**Ideal picture of future self/loved-one**

 Patient 3: *I'd just really like a chance at being...what everyone takes for granted. Just being a regular person with a regular job.*

 Caregiver 3: *I suppose just being able to live some form of a normal life and a happy life, or a happier life...[for her] to not have that terrible empty feeling all the time...Yeah, just to have somebody who is not always on the verge of suicide.*

 Caregiver 2: *I don't want her to be dependent on anyone. I don't want her to depend on me. I want her to be a fully independent person, making her own money, doing her own things. She doesn't have any of that. That would be good but that's the Holy Grail.*

 Caregiver 6: *I would love to have a bit of my old Christopher back [laughs]. I mean, look, I'll go for 100 per cent, I'll settle for 50, even 25. I mean, just so we go home and enjoy life together.*

Caregivers expressed hope that the person they had known prior to depression would be restored, but acknowledged that after long-term mental illness they were unlikely to be exactly the same. Caregivers also identified qualities they valued in their loved-one that they hoped would remain after DBS, such as being a loyal partner and parent, a gentle and caring person, and sharing a close bond. However, some were willing to lose these if it meant the patient could achieve relief from depression and improved quality of life. When imagining an ideal picture of their future self or loved-one, participants desired normality and the ability to simply "do things". Some patients envisaged achieving professional goals or travelling, while others wished to feel more confident and at ease amongst people.

With DBS, patients' perspectives of themselves were greatly influenced by the procedure's perceived benefit. Those who experienced a subjectively meaningful antidepressant benefit (i.e., described a pronounced improvement in mood that was sustained over multiple months) conveyed a sense of restored self (*n* = 2) (Table 2). Those who experienced modest or transient benefits (*n* = 3) (i.e., described subtle improvements in mood and/or notable lifts lasting only 2–3 days duration) recognised encouraging moments of their prior self (e.g., increased curiosity, interest in work activities). Caregivers recognised aspects of their loved-one previously, but were less emphatic about this restoration, noting the persistence of qualities that had developed with depression (e.g., sympathy-seeking, self-focused).

Decreased depression was accompanied by some transformation in self-concept; however, patients remained aware of the substantial distance between their idealised self (e.g., physical

**Table 2. Impact of mental illness and treatment on self-concept–Post-DBS.**

**Experiences of former self and personality**

Caregiver 2: *She's more interactive and more content and happy with things, so the kind of behaviours I'm seeing are what I tend to remember she was like before she had the depression. . .So, she's back, more back to her old self, I guess. . .[She's] started to feel more independent and wanting to do her own things, and plans for the future.*

Patient 4: *I don't think it's changed me, but there are little sparks of who I used to be coming back. Because I'm actually quite an outgoing person, so there's sparks of that coming back, instead of this awful person that would just sit. . .like [I] didn't even speak to people, I was so depressed. Couldn't look people in the eye, I couldn't even speak.*

Patient 6: *I'm back to where I was. No, I'm definitely back. I am the Christopher Daniels I was, dare I say it, 20 years ago. I'm not living in fear of being depressed. I'm not living in fear of being not in control, and being the victim. . .It's like looking in the mirror for the first time and seeing this person looking back at you. That person that you remember from 20 years ago.*

**Transforming self-concept and associated adjustment difficulties**

Patient 2: *Trying to describe what it felt like to be depressed and what it feels like now. There's no word for it. . .It feels like–the devil is when I was depressed, and I feel like an angel. . .I'd like to keep this personality, rather than being–I mean, I'm still nasty, angry at Doug [caregiver], because he's the only one that I'm usually in contact with, so if I have to take anything out, I usually take it out on him.*

Caregiver 2: *She's doing all these things, she's starting to feel better about herself, and she's hoping people see that and acknowledge that, and not everyone does. Everyone's involved in their own worlds as well, so she just has trouble expressing it, I guess.*

Caregiver 6: *He's different to how he was when he was grossly ill, but [he's] more like the Christopher that I had, just a little bit more self-centred. . .I think it's been all about him for so long. . .He has been ill a long time, and I assume it affects anybody, if you've been ill a long time. . .It's hard for him.*

**Fundamentally unchanged, transient experiences**

Caregiver 3: *I would say that there have been times where she's told me that she doesn't feel like herself, where she felt that she's done things that were out of character for her. . .There have been times where she has been experimenting with the DBS and those have led to the more manic episodes.*

Caregiver 5: *I think I've forgotten what the real Fiona is. . .it's not the real Fiona. . .That's not because of the DBS, but just in general.*

appearance, having friends, working etc.) and actual self. Novel qualities, including irritability and anger ($n = 3$) were reported by patients and caregivers. These were considered reactions to either the challenges of social reintegration, the procedure's perceived lack of effect, or a side-effect of a particular stimulation setting.

Patients who experienced no benefit felt fundamentally unchanged by DBS ($n = 2$), but transient experiences of acting uncharacteristically and feeling unlike oneself were reported in the context of stimulation-induced manic episodes ($n = 1$).

## Device acceptability and usability

In preparation for DBS, patients considered the prospect of having an implanted device, their relationship with it, and how this may affect their view of themselves and their body (Table 3). Most anticipated it having little impact and being easily adapted into their self-view. With the device encased within the body and not directly visible to others, most patients predicted being rarely conscious of it. Many likened it to other implanted devices for medical conditions, such as pace-makers and insulin pumps. Unlike some of these static devices, one patient highlighted the dynamic relationship with DBS, requiring setting adjustments and battery recharging. Another patient expressed mild disgust imagining electrodes embedded in their brain and the IPG in their chest. They anticipated feeling physically self-conscious, especially while recovering with a shaved head and scarring. Whether the intervention was successful was considered likely to dictate how the device was viewed (i.e., with pride or disgust).

Participants had considered implications of having a DBS device, including not flying after surgery, navigating airport security scanners, and needing reliable access to electricity, none of which were considered particularly restrictive. Participants were expecting to receive a rechargeable battery (to increase battery longevity) and were aware of the need to regularly

**Table 3. Device acceptability and usability–Pre-DBS.**

**Anticipations for implantation**

Interviewer: *How do you feel about the prospect of having a stimulator as part of you and as part of your body*?

Patient 3: [. . .] *It doesn't change how I identify myself. The only difference it will mean is, I have to go through a different line at the security—at the airport. . .If you break your arm and you get a plate put in, you don't suddenly think you're Iron Man and your whole sense of identity changes. . .I mean, for some things, yes, it does change you a bit. Like having a colostomy bag, that is a difficult change to adjust to and adapt. But because this is embedded in you and you can't see it–you might have to adjust it from time to time, but it's not going to cause discomfort or embarrassment. No one will know that you have one. It's really a non-issue.*

Patient 4: *Yeah, I feel that it's going to feel a bit creepy having it in my chest and knowing it's in my head. Especially until my hair grows back, I think I'm going to feel a bit yucky. Yeah. It's no different to having a cardiac pacemaker. . .I'm trying to think of it in those terms, but I wouldn't blindly say that I'm looking forward to having the stuff in my head and in my chest. . .I think maybe I'll look at that lump in my chest and. . .[if] I'm feeling much less depressed, I'm feeling happy again, I might look at it and go 'I think this is the best'–I'll wear this with pride and courage, wear this as a badge of honour because it's made me better. I think if I was mentally well, I would have a totally different view of this thing in my body, but the way I look at it now I go 'yuck', it's going to be a bit yucky.*

Interviewer: *Do you think. . .having the DBS, that it could change how Sarah views herself or sees herself*?

Caregiver 4: *Possibly, yeah, but I think that's part of the depression. I don't think it will change her and the way she sees herself as part of, well, I'm suddenly different now because I had DBS. I think it's more of I'm suddenly different now because I'm not ill anymore. No. . .we don't view the DBS as being any different to someone with a pacemaker. It's just another apparatus that helps you survive.*

**Device management**

Patient 1: *You have to have a reliable supply of electricity in order to continue to charge the stimulator which could have implications if you wanted to go and live in the Amazon for six months but I didn't weight that very heavily because it's not something in which I have any interest at all.*

Patient 4: *What happens in 30 years when all these doctors are dead and who's going to fix it then? I just have these thoughts about who's going to maintain me when these doctors retire?*

Patient 6: *I think I've got a reasonably technical. . .background to know where I draw the limit. I would certainly fully expect that I'd be trained over a period of time [to use it]. . .The only piece of equipment that I've seen were the stems with the two electrodes on the end. . .What I haven't seen is. . .the controller and the battery pack. . .Somehow you sort of wear a collar. . .It charges up–I don't know– 20, 30, 40 minutes and it holds the charge for three or four days. So, it's not really a great deal.*

Caregiver 4: *[The recharging] we've certainly been told. . .it's relatively simple. The recharger is only a little thing. You can hide it on your chest and it's an induction type recharge so we're not expecting anything unusual or unrealistic. In this modern era, we walk around with computers, phones and their chargers so it's almost part of current life.*

recharge. Some compared this process with charging a mobile phone–an everyday part of life. Those comfortable with technology felt confident they could learn to manage the device independently. Less confident patients expressed fear of "pressing the wrong button" and wanted control left in the professionals' hands. The need for device maintenance and specialist care in the distant future played on some patients' minds.

After surgery, most patients felt conscious of their physical appearance (shaved head, bandages, scarring) and how others perceived them (Table 4). For one patient, only once the stimulation was turned on and exerting some effect did they become conscious of their appearance. Patients avoided discussing the procedure or its purpose with others unless asked directly ("I just don't want people to think I'm a freak"). Some experienced pain, discomfort, and tightness around their IPG and subcutaneous wiring in certain positions or postures. For some, comfort and freedom of movement came with time, while others made adaptations to alleviate device-related discomfort (sleeping on one side, cushioning pillows). Patients tended to see themselves no differently with DBS and accepted the device as part of their body. For some, this acceptance had taken time with their perspective influenced by whether they believed DBS was having an effect.

One aspect all patients struggled with to varying degrees was battery recharging. The recharging experience had been one of frustration and annoyance, even for those who considered themselves easy-going or tolerant. Frustration was associated with difficulty establishing a reliable connection, inconsistency in how long and often recharging was required, recharging

**Table 4. Device acceptability and usability–Post-DBS.**

**Relationship between body and device**

Interviewer: *Do you think you've adjusted to having the actual device as part of your body*?

Patient 4: *I'm finding it a pain with the charge up, because now it can take two or three hours every second night. . .I'm thinking is this going to be an absolute bind for the rest of my life, trying to get this charging right. . .It's becoming quite ugh this is going to be really tough. It's going to prioritise my life.*

Patient 2: *After DBS. . .before I had it turned on; I didn't care about what I looked like. . .after the surgery, I went to the shops with all the bandages on my head–and being bald and everything–and not worrying about what I looked like. Whereas, once it was turned on, I started to worry about what it actually looks like, and everyone–how they thought it looked like as well. . .but as my hair did grow, I felt a lot more comfortable.*

Patient 6: *Yeah, I did go through a stage. I felt I was a–what's the word [pause] I felt I was like a monstrosity with this thing. I went through a phase where–I've let my hair grow longer, but you can still see–when you look in the mirror it's very apparent to you that you've had this procedure. You've got this thing floating around in your chest, but I'm through that.*

**Recharging experience**

Interviewer: *Do you think that she's [patient] adjusted to having the device as part of her body*?

Caregiver 3: *Mostly. . .one of the things is that it's taking her a very, very long time to charge it, so that's very restrictive. . .so I think that side of it is frustrating. One of the hopes was that–obviously having ECT. . .you're having a general anaesthetic, you can't drive–it's affecting your lifestyle. So, the hope was with the DBS that it's. . .not having to go through all of that, but while you're having to be plugged into the wall for hours at a time, it does affect your lifestyle. So, I suppose from that side of things, she's not accepting it as part of her body because she's finding that part unacceptable.*

Caregiver 2: *She was getting very frustrated with that [recharging] because sometimes it takes longer to charge than others. . .Yes, I guess the reality is always different to what you're expecting, but she's getting used to it. That will always be a little bit of a burden I guess because now everywhere we go she needs to take the stuff with her, and make sure she's got it with her. I guess it's just not that much different to people that take heart medication. . .So, if we were to go on a holiday somewhere, we have to make sure we've got power sources to charge it up, just things like that. It is an annoying factor, but considering what benefit it brings. . .*

Caregiver 4: *Having the implant I don't think has really impacted it in terms of from a personality thing or from a view of herself. I think she is quite comfortable with that. . .It's just the physical bit; the recharge is quite a challenge and it's quite time consuming. . .Every other day it's a couple of hours. . .so if you don't do it for two days, its battery life is quite depleted. . .So that's her biggest challenge at the moment, but I think she is quite accepting of the fact that if I feel better and that's what I have to do, that's what I do. She says it's no different to someone having dialysis every other day or every third day. So, it's just an accepted part of the illness.*

**Device control**

Patient 4: *I just feel nervous when I'm putting it up, because I'm scared I'm going to push the wrong button and switch it off or do something. . .I'd rather my husband did it, because I get nervous about doing it. I'm much more familiar with it now and I'm much better. . .[But], he's not so upset when he's doing it or nervous or depressed. He's in a good place.*

Caregiver 4: *No, the device is simple. No. Sarah has, and again, I think because of depression, she doesn't take it all in. . .[It's] relatively straight forward, [the DBS clinician] will say to us, 'you can go up a setting' or whatever the number is. It's quite easy. That's very easy to do. . .[And] not that Sarah ever wants to turn it off, but if you want to turn it off, you know how to. It's quite simple.*

Caregiver 5: *There was one episode. . .where she [patient]. . .just couldn't bear herself. . .Her anxiety about readjusting down was difficult for her, but she eventually. . .got to a stage that I had to do something so yeah, we turned it down and that reduced it down. . .I try to leave it for Fiona as much as I can, but. . .unfortunately [she] is not very good with technology and. . .having the depression all these years.*

taking longer than originally expected, and the growing need to prioritise recharging in daily life. Recharging requirements varied depending on each individuals' settings (greater recharging burden associated with higher voltages). Some were still establishing an optimal recharging routine. Many had made contact with their device company representative to receive reassurance and guidance around recharging technique (e.g., understanding how body temperature, posture, and tension influence connection).

Within the open-label phase, arrangements for adjustment of stimulation settings varied for each individual. They were influenced by the patient's geographical location, accessibility of support people (caregiver, psychiatrist), size of the intended adjustment, plus the patient's preference. Patients' level of comfort controlling the device (turning on/off, small setting adjustments) were consistent with their pre-DBS comments around technological confidence. Some preferred all adjustments be made by the DBS specialist, but this was not always possible

or practical, due to geographical distance. This occasionally required patients to make adjustments with caregiver or psychiatrist support. One patient with no caregiver had all adjustments made while in hospital. They described experiencing cognitive side-effects with one setting and how these effects would have impacted their ability to re-adjust the settings independently. Outside the specialist clinical trial team, participants' experience with medical professionals broadly was that DBS, particularly for depression, was not well-understood. Device company representatives who provided patients with device education peri-operatively, were also engaged to guide medical professionals around appropriate device management during medical procedures unrelated to DBS.

## Relationships and connection

In pre-surgery interviews, patients and caregivers discussed how their relationship had been impacted by depression (Table 5). Those in spousal relationships had experienced a shift in dynamics, with caregivers performing and identifying with the 'carer' role to varying degrees ("pseudo" to "full-time"). All expressed their commitment to and support of one another, but acknowledged their relationship was not reflective of a true husband and wife relationship. Couples described their inability to plan due to the uncertainty of the illness. Plans for socialising or holidays were avoided, with couples living a day-to-day existence and keeping their world small (family, close friends only). Caregivers had freedom to do activities independently, but expressed their preference to share and enjoy these experiences with their partner. In addition to intense sadness and suicidal ideation, some patients experienced moments of aggression and intolerance, which could be directed towards caregivers. A decrease in open, honest communication was described. Some caregivers were "walking on eggshells", while some patients withheld details about the depths of their depression or suicidal thoughts. In certain cases, suicide attempts or completions had been prevented by caregivers, either in a physical sense or due to patients not wanting to cause them anguish. Despite difficulties posed by depression, patients and caregivers maintained strong and meaningful bonds. For patients with children, discomfort was associated with their children seeing them unwell and patients avoided allowing them too much insight into their illness. Social relationships more broadly were considered effortful to maintain and the prospect of social interaction was anxiety-provoking for most patients.

Both patients and caregivers hoped aspects of their relationship would change after DBS. Patients hoped caregivers would not have to regularly bear witness to their intense sadness or anger. Some felt indebted to their caregiver and hoped to be able to repay them. Others desired greater physical intimacy with their spouse and hoped DBS would improve their sex drive. Caregivers hoped for a more equal relationship and to be able to enjoy "small things, but meaningful things" together (e.g., having a coffee, going for a walk). Many caregivers experienced worry and guilt when leaving the patient at home alone for fear of what they would return to and hoped DBS would alleviate this heavy burden. All participants wished for greater ease when interacting with others and that time spent with family and friends would be enjoyed rather than endured.

Some participants reflected on how society tends to respond to individuals with mental illness. It was felt that people distance themselves when depression is involved and rarely show the same compassion and support to families that they would if it were a different illness, such as cancer.

Post-DBS, those who had experienced notable benefits described how their relationships had both improved and become more complicated (Table 6). Changing dynamics and re-negotiating roles occasionally led to increased conflict (e.g., patient more interactive and

**Table 5. Relationships and connection–Pre-DBS.**

**Impact of depression on relationships**

Interviewer: *What changed for you in the way that you live since Deborah got depression?*

Caregiver 2: *Well it's very much multiple aspects, because from a relationship point of view that changes completely because you're not–well, you're still husband and wife but she just relies on me for everything to the point where she says she wouldn't be alive if I wasn't around [laughs] it's a pretty big responsibility. It's not something you really want. You're supposed to be there as a team and challenging each other and supporting each other but it's all very one sided. So, from that point of view it's a very different relationship than what we had before.*

Patient 1: *Depression itself is incredibly corrosive. Incredibly corrosive of relationships generally so I suppose were I to obtain some benefit, there might be some improvement in that area of my life. . .As things are right now, it's incredibly effortful to maintain relationships at all. It's really difficult if people ring me, just even to answer the phone. . .It's difficult to spend time with others, it's just exhausting. I would say certainly from the treatment at that level, it would just be a bit less effortful for me and I think that would have implications in terms of the quality of relationships as well.*

Caregiver 4: *You can look at yourself as a pseudo carer and just someone that's tagging along. We're there, we're together, but there's not really a proper relationship if you like. . .and that has to improve because it's not much of a life for either of us. . .There's still a commitment and you want to help someone get better but it's pretty tough for both of us. . .I think that's just where we want to go.*

Caregiver 6: *He loves his family. . .he loves but he can't show any warmth or affection. . .He's so preoccupied with. . .his illness and he's missing so much and he knows he is missing so much. . .It robs you of all these things.*

**Hopes for future relationships**

Interviewer: *How about yourself? How would you like your own picture to look in a year's time?*

Caregiver 4: *Oh, for me, I mean I'm just happy if–I'm happy when she's happy and I'm happy when I can walk out the door and know that she's safe. . .I'd like to do things together. . .but, at the moment we can't, we don't. . .So, for me it's just hoping to return to where we were. . .where we included our friends and we integrated more with people–and ourselves.*

Caregiver 2: *I just want to be in an open and honest relationship again–it's not a dishonest one now, you just have to filter what you say–and just communicate our wants and desires out of life. Just on a more equal footing in the relationship, that's what I'm expecting. . .Our relationship as husband and wife. . .how do we start re-engaging again? It might be hard for me to let go of making all the decisions or controlling everything. After ten years of it you start getting used to it, I suppose. Yeah, I guess we expect that. . .if she starts getting better our relationship will change again and we'll probably need some guidance for that to help us get through that. We'll see how we go.*

Patient 5: *I feel my [children], I'm not quite sure how to put this, see me as sick in some way, and I'd really like that to not be the case. I want them to see me as a fully functioning person, whereas, yes at the moment I just feel they see me as someone that's sick.*

**Society and depression**

Patient 3: *That's one of the biggest things that a lot of people don't seem to understand is that when you have depression it doesn't just impact you; it impacts the people who are close to you. It takes a huge toll on them. . .They seem to understand, okay, if someone is dying of cancer, that they're going through chemo–it's very emotional, it's very hard and they give the. . .carers a bit more love and empathy. But it's not the same with depression. . .They'll wait until that person has turned blue, and then suddenly all put on their little sympathy hats.*

Patient 4: *You get a lot of sympathy if you get cancer for eight years but when you've had depression for eight years people just are a bit sick of it really. If you've got cancer, they're having a fundraising ball for you but depression is actually probably worse than cancer because with cancer you live or die. Depression is a life sentence if you don't get rid of it.*

Caregiver 3: *Unfortunately, a lot of the times when people do have a mental illness, people do try, tend to treat you like you've got no brains, type thing.*

independent, caregiver needing to relinquish control). Despite improvements, some dynamics remained ingrained (e.g., patient not contributing to housework, being waited on by caregiver). Patients' increased energy and motivation to do things was generally considered beneficial for the couple. However, caregivers occasionally tempered patients' impatience and enthusiasm to jump back into things quickly (e.g., overseas travel, return to work). These appeared driven by patients' desire to make up for lost time or re-establish their independence and identity. Caregivers felt able to be more open and expressive in their communication, although would still withhold or filter certain information. Patients' risk of self-harm had reduced, which was a considerable relief for caregivers. Relationships with family tended to improve, with patients more engaged and nurturing with children and grandchildren. Patients expressed interest in developing or re-establishing friendships, but were also hesitant due to

**Table 6. Relationships and connection–Post-DBS.**

**Wellness and adjusting relationship dynamics**

Interviewer: *Would you say that your relationship with Clementine [caregiver] has changed*?

Patient 6: *Oh yes. It has to have been. I've gone from–I've been a dead duck to–we're more involved in each other's lives now...I think last week we were out three times...I'm just looking forward to life being a hell of a lot better than what it has been...I'm glad to say my family have rolled with the punches. They've been there on the bad days and they're going to be there for the good days now...Let me put it this way, I feel I'm closer to my family than anything I've known for a long time, because there's just been so many occasions...I haven't been able to go because I've been too unwell.*

Interviewer: *Do you and Deborah feel like different people now?*

Caregiver 2: *I don't think so–well, no, yeah, she does definitely. Yeah, I don't know, it's a hard question that one. She definitely, for sure, and I guess I'm still trying to figure out where my place in all this is really, and still doing all the things that you do, but our relationship is obviously going to change as time goes on. We're getting there, but it doesn't come without its own set of drama.*

Interviewer: *If you were to describe your relationship since DBS, would you say things are better, easier, more strained, a bit more complex?*

Caregiver 2: *It's more complex actually, which is interesting, when you think it wouldn't be, but I think really when someone is so depressed and so low all the time you just get into that routine, and your interactions are pretty, day-by-day, the same...So, now she's become her own self, interactive and all that sort of stuff, so there's definitely more complexity to it for sure. I think some of...those roles that I was performing, which I need to let go of, these habits for me that I have to let go of. I'm not perfect, and sometimes I don't take that kind of feedback well [laughs]...[It's] definitely more complicated, but I think overall, she's certainly better...and she can express herself a bit better. Plus, I can express myself a bit more, which prior, it'll still be this way, but before I couldn't have a bad day...At least now there's a bit more scope to be able to share those own stresses that I have in my life. But it's still early days, and I still have to choose the amount information and the colour of it. But I think as time goes on it'll get–it'll all interact more freely like a couple should...We're certainly getting back more to that, and more of an equal relationship. It's not quick, it's going to take time.*

Caregiver 6: *Well, it's not just an improvement in his mental health, it's an improvement all around...I would have to say, because his mental health at the moment is pretty good, it [the relationship] has to be easier. I'm not saying it's perfect, but it has to be easier. Because–yes, I've got less worries of him trying to harm himself...He's very pedantic about things, and I don't see eye to eye with, but if I question it, I'm wrong. So, I'm learning somewhere along the line what to question and what to not.*

**Impact of no benefit on relationships**

Patient 4: *Parents are not meant to be depressed, they're meant to be your rock, and when that rock gets depressed and is sobbing in a corner, that's very challenging as a child I imagine.*

Patient 5: *I've probably said more to my [children]. Sometimes they'll say 'sometimes you say you're all right but you're not'. I suppose I have been a bit more open with them.*

**Caregiver wellbeing and support**

Caregiver 5: *I've got a couple of good mates, we talk things through and always checking in how things are going...So, that's been pretty good. I know I can talk to my [children] about it...but I haven't gone, haven't sought any professional help at this stage. Sometimes I think I should...Yeah, because I mean sometimes it does wear you down– then you think what she [patient] is going through, if I'm only going through this [laughs].*

Caregiver 6: *I belonged to a group once, where you used to meet up and talk, and that was fairly good, because you gave each other support on different things. That sort of petered and fell apart...I rang the council to see what they had...there isn't a lot, really, of support, I have to say. It's–there's a fair bit of stuff that the ill person can do, but the people who look after them, there isn't a lot. Which is sad. There must be so many people like me.*

out-of-practice social skills and fear of judgement. Mental health support groups had been considered as avenues for building social connections.

For those who experienced little-to-no benefit from DBS, their relationships remained mostly unchanged and functioned as before. However, the process of undergoing surgery and participating in the trial resulted in some changes to relationship quality. For one patient, the time and energy they had available for others prior to DBS was depleted through the process of regularly travelling interstate for stimulation adjustments. This was difficult for some friends to understand and accept, resulting in more strained relationships. The tendency for some patients to display irritability after DBS led to caregivers becoming more irritable themselves. Instances of stimulation-related adverse events (e.g., mania, suicidal ideation), had been confronting experiences for caregivers and close-others and left lasting impressions. The experimental procedure had prompted more discussions about depression amongst families (including children), but patients still avoided the topic so as not to burden others.

Caregivers discussed the relationships they had available to support their own wellbeing. Most caregivers had friends they could speak openly with or adult children they could seek support from. Some caregivers took antidepressant medication to help manage their daily stresses but none were engaged in psychotherapy, although a number felt perhaps they should be. A lack of support available to carers of people with mental illness, both in society generally and within community services (e.g., support groups), was noted.

## Discussion

The aim of the current study was to qualitatively examine how DBS for TRD impacts patient personality, self-concept, and relationships, from both patient and caregiver perspectives. These narrative, first-hand accounts add to our understanding of DBS and provide valuable insights that are typically not captured in randomised-controlled trials. We discuss these findings in light of existing DBS literature and consider their clinical implications.

### Personality and identity in treatment-resistant depression

Clinical and ethical concerns have been voiced about the potential for neurointerventions such as DBS to result in personality change [30, 31], yet in the case of psychiatric disorders some researchers and clinicians argue this is the intended outcome [32, 33]. Synofzik and Schlaepfer [34] suggest that if personality–that is "a dynamic and organised set of characteristics in a person that uniquely influences his or her cognitions, motivations, and behaviours in various situation" (p. 1514)–remains unchanged, the intervention has failed. The hope many have, including the current participants, is that DBS will restore premorbid personality and self. This is a reasonable desire; however, it should not be expected that individuals who have endured years or decades of depression (as most patients had) will go completely unchanged by this experience and be restored unmarked [35]. In the present sample, onset of depression for most was during adulthood (*n* = 5) after they had established distinct identities and personalities. The degree to which patients felt their old personality or self had been restored was closely associated with the degree of benefit they experienced with DBS. Still, even patients who felt much restored identified ways in which they were far from who they were previously, particularly regarding functional capacity to perform meaningful activities and roles (working, travelling, socialising). Similarly, caregivers recognised elements of their loved-one previously, but continued to see persistent illness traits (e.g., sympathy-seeking, self-focus).

Previous DBS clinical trials have investigated the relationship between antidepressant effect and personality, as measured using the NEO Five-Factor Inventory (NEO-FFI). Psychometric personality scales, such as the NEO-FFI, often appear homogenous from individuals with severe depression. They are characterised by high Neuroticism and low Extraversion, regardless of age [36]. These profiles tend to reflect the nature of the illness more than the individual themselves [37], suggesting that if depression is alleviated through DBS it should result in a change in NEO-FFI scales. Bewernick et al. [38] found no difference in Five-Factor scales at 6-months, 2 years or 5 years after DBS of the supero-lateral branch of the medial forebrain bundle (*n* = 21), despite patients obtaining an antidepressant response. The authors suggest this may reflect a 'scar effect' in which lengthy depressive episodes result in long-term, irreversible changes to personality. In contrast, Ramasubbu et al. [37] found a significant decrease in Neuroticism and increase in Extraversion following DBS of the subcallosal cingulate (*n* = 22), with patients assessed regularly up to 15-months. The personality changes were associated with clinical improvement in depression, with the direction of these changes aligning with normative personality data. The contrasting outcomes across the two studies may be explained by differences in target nuclei, assessment timelines, and patient populations.

Results from the current study could lend support to either finding, given those who experienced antidepressant effect described a sense of a restored personality, yet the lingering depressive qualities support Bewernick et al.'s suggestion of a 'scar effect'.

While an exact restoration of premorbid personality or self is improbable, if not impossible, they serve as helpful guideposts for recovery. The process is more complex where mental illness has emerged in adolescence, during important stages of personality and identity development [39, 40]. This is often the case in OCD and regularly occurs in TRD. Following DBS, rather than attempting to pick up where one left off, these patients must attempt to forge a new identity and personality that is not structured around mental illness [33, 41]. Identity challenges and poor psychosocial adjustment seen post-operatively in neurological populations has been described as the *burden of normality* [42, 43] and has been applied in psychiatric DBS cases [44]. This phenomenon where patients and caregivers experience difficulties transitioning from chronically ill to suddenly well, highlights the need for preparatory work and psychosocial rehabilitation in DBS to assist with regaining wellness and transitioning self-concept. Not all of the current patients were engaged in psychotherapy after DBS, but for one patient who gained benefit from DBS this support was vital for their psychosocial adjustment. In OCD samples, cognitive behavioural therapy (CBT) has been shown to augment DBS treatment effects [45] and is recommended for best-practice post-operative care [46]. In TRD samples, there is no evidence indicating that CBT augments the DBS treatment effect [47]; however, psychological therapy is likely to play an important role in relapse prevention. DBS is only a starting point to living well, as the scaffolding around the patient will still reflect someone with a chronic and severe mental illness (e.g., limited social connections, ingrained dependency on caregiver).

## Relationship with and acceptance of the device

Patients did not experience sustained fundamental changes in self-perception or feel dehumanised by the implanted electrical device, as has been reported in some DBS studies [24]. However, certain factors impacted patients' relationship with and acceptance of the device. These included: the noticeability of scarring, intermittent pain associated with IPG and wiring, DBS' perceived benefit (or lack of), and recharging needs. Recharging was a frequently difficult and frustrating experience. If an implanted device functions well, it typically goes unnoticed and is easily incorporated into one's body schema [48]. However, if it requires frequent attention and creates frustration (through regular, lengthy recharging), this impacts agency and may reinforce internalised feelings of defectiveness, particularly if no therapeutic benefit is gained. Patient satisfaction with rechargeable DBS products has been evaluated, but primarily in movement disorders [49, 50]. The recharging needs for individuals with psychiatric conditions typically differs from those with movement disorders (higher voltages resulting in faster battery depletion), as does their psychological and cognitive profiles. In OCD, post-operative education has been shown to assist with device-related anxiety [51], but further investigation is needed. Notable memory difficulties, which the majority of current patients reported (*n* = 4), can affect retention of device-related information. For this reason, perioperative device education may require modification for TRD purposes. This may include consideration of when device education is offered (e.g., not immediately after surgery) and conducting multiple sessions. Those who had received support from their device company representative in the months following surgery had found this helpful for improving their recharging process (achieving a more reliable recharging connection). A reduction in how long and how often recharging is required would increase the tolerability of the process. This is dependent on the capacity of the current technology. As DBS for psychiatric indications progresses, so too should device technology and research evaluations of patients' device-related needs [17].

## Relational adjustment

For patients who experienced therapeutic benefit, renegotiating the patient-caregiver relationship to a more egalitarian dynamic was a challenge. Even positive changes (e.g., patient exerting independence, eager for daily outings) were occasionally difficult for caregivers, as they broke predictable patterns. In De Haan and colleagues' [52] OCD sample, patients reported that both they and their partners had to "get used to" the new them. As OCD had always been present in their relationship, couples had created lives that worked around the illness. For the married couples in the current sample (n = 4), their relationships were established well before the onset of depression. Therefore, partners were experiencing varying degrees of 'reacquaintance' with their loved one (depending on benefit gained from DBS). Regardless of outcome, DBS created challenges for marital relationships. These couples, however, had weathered similar storms across their long-term, committed relationships. So too was the case for the parent-child pair.

The caregiver participants in the study possessed valuable knowledge of the patient and were well-positioned to provide observational feedback. This is one reason caregivers should routinely be involved in patients' DBS clinical care. Caregivers should also be included in clinical research to assess the impact DBS has on their own lives. DBS can be a transformative experience involving considerable adjustment for both parties [53]. In PD, there is growing recognition of caregivers' vital role in patients' DBS management, as well as the procedure's impact on their own wellbeing. A lack of support to assist caregivers with the implications of DBS has been noted. A number of pilot studies have attempted to address this issue, including an 8-session psychoeducation program designed for patients and caregivers [54], and an 8-session individual CBT program for caregivers post-DBS [55]. Although small samples, these promising interventions contribute to an area of unmet need. Given the unique challenges created by DBS, programs designed specifically to meet these needs are recommended to support caregiver wellbeing.

## Limitations

The patient sample (n = 6) was small and reflects the limited number of patients undergoing DBS for TRD in Australia. When collecting in-depth, longitudinal qualitative information, samples of this size can be sufficient and data saturation was achieved in the analysis of pre- and post-DBS data [56, 57]. The triangulation of caregiver perspective, in addition to multiple time points, added to the richness and depth of the analysis. Qualitative research strives for naturalistic generalisability (e.g., resonance with personal experiences and observations) rather than statistical-probabilistic generalisability, and is achieved through the select presentation of rich and in-depth subjective data [58]. It remains important, however, to acknowledge the influence of specific contextual features (e.g., target location, lead placement, research trial protocol, patient characteristics, geographical location) upon the experiences and outcomes described. While a mixed-method approach could have produced findings with both naturalistic and statistical-probabilistic generalisability, this approach was not feasible as the small experimental sample prohibited the conduct of robust quantitative analyses on psychometric data. These patients were extreme in depression and treatment resistance compared with others who have undergone DBS for depression globally. The majority of the patient sample were female (5 of 6), which may reflect the higher representation of females in depressed populations. This gender imbalance is likely to have influenced the nature of the observed findings, potentially reducing the transferability of these findings to males with depression. Few patients experienced sustained meaningful benefit at the time of follow-up (n = 2); therefore, our insights into the process of adjusting to wellness are from limited cases. There is a crucial

obligation, however, to learn as much as we can from the few DBS for TRD cases that exist, regardless of outcome [59]. Repeat interviews in subsequent years when patients have had more time with the device would have been valuable to see how their experience evolved over time.

## Conclusions

This is the first qualitative study of its kind to provide in-depth insight into the lived experience of DBS for TRD, including perspectives from patients and their closest supports who are vital throughout the preparation and recovery process. Change in self-concept was highly linked to therapeutic response and was part of a larger process of adjustment for patients, caregivers, and their relationship. Caregiver-specific support is recommended for managing this challenging process. Patients' relationship with the implanted device was marked by recharging challenges. Depression-specific research on patient satisfaction and user experience with rechargeable IPGs is required.

## Supporting information

**S1 File. Interview schedules for patients and caregivers.**
(DOCX)

## Acknowledgments

The authors wish to thank all the participants for sharing their experiences.

## Author Contributions

**Conceptualization:** Cassandra J. Thomson, Rebecca A. Segrave, Eric Racine, Adrian Carter.

**Data curation:** Cassandra J. Thomson.

**Formal analysis:** Cassandra J. Thomson, Rebecca A. Segrave, Adrian Carter.

**Investigation:** Cassandra J. Thomson.

**Methodology:** Cassandra J. Thomson, Rebecca A. Segrave, Eric Racine, Adrian Carter.

**Project administration:** Cassandra J. Thomson, Karyn E. Richardson.

**Resources:** Paul B. Fitzgerald.

**Supervision:** Rebecca A. Segrave, Adrian Carter.

**Writing – original draft:** Cassandra J. Thomson.

**Writing – review & editing:** Cassandra J. Thomson, Rebecca A. Segrave, Paul B. Fitzgerald, Karyn E. Richardson, Eric Racine, Adrian Carter.

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
