## [Decision Letter · Decision Letter 0]

30 May 2022

PONE-D-21-34707Personal and relational changes following deep brain stimulation for treatment-resistant depression: A prospective qualitative study with patients and caregiversPLOS ONE

Dear Dr. Thomson,

Thank you for submitting your manuscript to PLOS ONE. After careful consideration, we feel that it has merit but does not fully meet PLOS ONE’s publication criteria as it currently stands. Therefore, we invite you to submit a revised version of the manuscript that addresses the points raised during the review process. Please note that we have only been able to secure a single reviewer to assess your manuscript. We are issuing a decision on your manuscript at this point to prevent further delays in the evaluation of your manuscript. Please be aware that the editor who handles your revised manuscript might find it necessary to invite additional reviewers to assess this work once the revised manuscript is submitted. However, we will aim to proceed on the basis of this single review if possible.  Your manuscript has been assessed by an expert reviewer, whose comments are appended to this letter. The reviewer has requested clarification on the relationship between this paper and a related publication, and made some suggestions for expanding your literature review and discussion. Please ensure you address all of the comments carefully when preparing your revised manuscript. 

We look forward to receiving your revised manuscript.

Kind regards,

Joseph Donlan

Editorial Office

PLOS ONE

Journal Requirements:

2. Thank you for stating the following in the Competing Interests/Financial Disclosure section:

“I have read the journal's policy and the authors of this manuscript have the following competing interests: PBF has received equipment for research from MagVenture A/S, Nexstim, Neuronetics and Brainsway Ltd and funding for research from Neuronetics. He is a founder of TMS Clinics Australia.’

We note that one or more of the authors are employed by a commercial company: MagVenture A/S, Nexstim, Neuronetics and Brainsway Ltd

Reviewers' comments:

Reviewer's Responses to Questions

**Comments to the Author**

1. Is the manuscript technically sound, and do the data support the conclusions?

Reviewer #1: Yes

2. Has the statistical analysis been performed appropriately and rigorously? 

Reviewer #1: N/A

3. Have the authors made all data underlying the findings in their manuscript fully available?

Reviewer #1: Yes

4. Is the manuscript presented in an intelligible fashion and written in standard English?

Reviewer #1: Yes

5. Review Comments to the Author

Reviewer #1: This study qualitatively examined the personality, identity and interpersonal changes following DBS of bed nucleus of stria terminalis in 6 patients with treatment resistant depression. The authors devised a semi structured interviews focusing on i) background on DBS, ii) expectations/perspectives, iii) personality/identity, iv) interpersonal relationships, v) informed consent and decision making, vi) stimulator management. The interviews were conducted with participants and family care givers pre-DBS and 9 months post-DBS.

Previously published paper by the same authors titled “ Nothing to lose, Absolutely everything to gain” ( Frontiers in Human Neuroscience Sep 2021) addressed two themes emerged from the questions on pre-DBS expectations/perspectives and post-DBS experiences and perspectives, pre- and post-DBS informed consent and decision making. The current paper focused on personality/ identity and interpersonal relationship.

Comments:

1. Although the results are presented in two parts in two papers, and mentioned as non-overlapping, there seem to be implicit overlap in conception, thematic analysis, and conclusions. Ideally these two papers should have been combined into one paper. Authors should clearly mention overlapping and non-overlapping components in methodology.

2. Personality and identity are complex multifaceted psychological concepts. Participant’s response may be influenced by their level of understanding and knowledge about personality and identity. Did the authors brief the participants about the dimensions of personality and identity during the interview?

3. Limitations of qualitative methods can be minimized by using mixed model. This need to be addressed in limitations

4. Minor errors.

a) Introduction Page 3 line 66. “There have been no studies investigating patients experience of DBS in TRD”. This should be changed to there have been no in-depth studies investigating patient’s experience. There were two studies that examined patients subjective experience along with objective personality changes in NEO. ( Bewernick et al 2018 Psychol Med ; Ramasubbu et al J 2021 Psychiatry Neurosci).

b) Discussion: Page 21; line 380-381. “ CBT has been recommended as a best practice component for DBS recovery and shown to augment treatment effects (43)”.

This should be changed to “ shown to augment treatment effects in obsessive compulsive disorder (43). Add on CBT has no effect on DBS recovery in treatment resistant depression (TRD) (Ramasubbu et al 2020 Lancet psychiatry).

6. PLOS authors have the option to publish the peer review history of their article (what does this mean?). If published, this will include your full peer review and any attached files.

Reviewer #1: No

---

## [Author Response · Author response to Decision Letter 0]

22 Aug 2022

Dear editor/reviewer, 

Thank you for taking the time to review our manuscript and for your constructive feedback. Responses to reviewer comments (in bold) and other manuscript amendments are listed below. Page/line numbers align with the revised version of the manuscript containing track changes. 

Reviewer #1: This study qualitatively examined the personality, identity and interpersonal changes following DBS of bed nucleus of stria terminalis in 6 patients with treatment resistant depression. The authors devised a semi structured interviews focusing on i) background on DBS, ii) expectations/perspectives, iii) personality/identity, iv) interpersonal relationships, v) informed consent and decision making, vi) stimulator management. The interviews were conducted with participants and family care givers pre-DBS and 9 months post-DBS. 

Previously published paper by the same authors titled “Nothing to lose, absolutely everything to gain” (Frontiers in Human Neuroscience Sep 2021) addressed two themes emerged from the questions on pre-DBS expectations/perspectives and post-DBS experiences and perspectives, pre- and post-DBS informed consent and decision making. The current paper focused on personality/ identity and interpersonal relationship.

Comments:

1. Although the results are presented in two parts in two papers, and mentioned as non-overlapping, there seem to be implicit overlap in conception, thematic analysis, and conclusions. Ideally these two papers should have been combined into one paper. Authors should clearly mention overlapping and non-overlapping components in methodology.

The authors considered at length the option of consolidating all the findings from the study into a single paper. Due to the complex nature of the qualitative data collected (that includes multiple participant groups, timepoints, and research questions of interest), we concluded that attempting to condense all of these findings into a single paper would not be possible within the constraints of all leading clinical and scientific journals. Inclusion of all of the findings within a single paper would also not do justice to the data. We believe that the two papers addressed two different clusters of themes that are more or less relevant to different populations of readers. The currently published paper deals with questions of responsible research in DBS trials, and of primary interest to clinical and scientific teams conducting trials, whereas this current study examines the impact of DBS trials on personality and interpersonal relationship that would be of relevance to a range of disciplines (e.g., social science, philosophy), and not just clinicians and scientists. Including all themes within a single paper would make it overly dense and largely incomprehensible. In order to remain within journal word limits and allow for sufficient exploration of themes, the decision was made to publish two separate papers presenting two cohesive clusters of themes that pertain to different research and interview questions. We agree that greater clarity in needed to distinguish the overlapping and non-overlapping components of the two papers: 

EDIT TO MS [Pg. 5 Line. 101]:

Separate data findings derived from these interviews, on the topics of informed consent, participant expectations and subjective patient outcomes, are reported elsewhere (27). 

2. Personality and identity are complex multifaceted psychological concepts. Participant’s response may be influenced by their level of understanding and knowledge about personality and identity. Did the authors brief the participants about the dimensions of personality and identity during the interview?

We deliberately chose not to provide participants with a definition of personality or identity or any briefing on these concepts. We were not interested in their views on scientific or psychological constructs of personality or identify. Rather we were curious to see how participants made sense of these concepts for themselves, and how this influenced their perspectives on their experience of DBS and depression. This point has been included in the manuscript. 

EDIT TO MS [Pg. 6 Line. 129]:

Pre-surgery interviews occurred 3 to 15-weeks prior to surgery (M = 46 minutes, range = 34-58 minutes) and explored participants’ knowledge of DBS and beliefs about how the patient’s personality, self and relationships may be affected (see Supplementary Material for interview schedules). Participants were not provided with a definition or description of ‘personality’. We were interested specifically in their personal experiences of personality and identity change. This was done to avoid restricting participants’ responses to a particular conceptual framework.

3. Limitations of qualitative methods can be minimized by using mixed model. This need to be addressed in limitations

EDIT TO MS [Pg. 24 Line. 470]:

While a mixed-method approach could have produced findings with both naturalistic and statistical-probabilistic generalisability, this approach was not feasible as the small experimental sample prohibited the conduct of robust quantitative analyses on psychometric data.

4. Minor errors.

a) Introduction Page 3 line 66. “There have been no studies investigating patients experience of DBS in TRD”. This should be changed to there have been no in-depth studies investigating patient’s experience. There were two studies that examined patients subjective experience along with objective personality changes in NEO (Bewernick et al 2018 Psychol Med; Ramasubbu et al J 2021 Psychiatry Neurosci).

This statement has been amended according to reviewer’s suggestion. The two studies have also been expanded upon in the discussion section. 

EDIT TO MS: [Pg. 3, Line. 69]

While the efficacy and safety of DBS for TRD continues to be investigated via clinical trials, with mixed results (13), there have been no in-depth studies investigating patients’ experience of DBS and their perspectives on the psychosocial changes they undergo.

EDIT TO MS: [Pg. 20, Line. 364]

Two studies have investigated the impact of DBS for TRD on personality profiles using the NEO Five-Factor Inventory. Bewernick et al. (32) found no difference in Five-Factor scales at 6-months, 2 years or 5 years after DBS of the supero-lateral branch of the medial forebrain bundle (n = 21), despite patients obtaining an antidepressant response. The authors suggest this may reflect a ‘scar effect’ in which lengthy depressive episodes result in long-term, irreversible changes to personality. In contrast, Ramasubba et al. (31) found a significant decrease in Neuroticism and increase in Extraversion following DBS of the subcallosal cingulate (n = 22), with patients assessed regularly up to 15-months. The personality changes were associated with clinical improvement in depression, with the direction of these changes aligning with normative personality data. The contrasting outcomes across the two studies may be explained by differences in target nuclei, assessment timelines, and patient populations. 

b) Discussion: Page 21; line 380-381. “CBT has been recommended as a best practice component for DBS recovery and shown to augment treatment effects (43)”. This should be changed to “shown to augment treatment effects in obsessive compulsive disorder (43). Add on CBT has no effect on DBS recovery in treatment resistant depression (TRD) (Ramasubbu et al 2020 Lancet psychiatry).

EDIT TO MS: [Pg. 22, Line. 410]

In OCD samples, cognitive behavioural therapy (CBT) has been shown to augment DBS treatment effects (43) and is recommended for best-practice post-operative care (44). In TRD samples, there is no evidence indicating CBT augments the DBS treatment effect (45); however, psychological therapy is likely to play an important role in relapse prevention.

---

## [Decision Letter · Decision Letter 1]

5 Dec 2022

PONE-D-21-34707R1Personal and relational changes following deep brain stimulation for treatment-resistant depression: A prospective qualitative study with patients and caregiversPLOS ONE

Dear Dr. Thomson,

Thank you for submitting your manuscript to PLOS ONE. After careful consideration, we feel that it has merit but does not fully meet PLOS ONE’s publication criteria as it currently stands. Therefore, we invite you to submit a revised version of the manuscript that addresses the points raised during the review process.

The manuscript has been evaluated by two reviewers, and their comments are available below.

The reviewers have made several requests for additional details and clarifications.

Could you please carefully revise the manuscript to address all comments raised?

We look forward to receiving your revised manuscript.

Kind regards,

Steve Zimmerman, PhD

Associate Editor, PLOS ONE

Journal Requirements:

Reviewers' comments:

Reviewer's Responses to Questions

**Comments to the Author**

1. If the authors have adequately addressed your comments raised in a previous round of review and you feel that this manuscript is now acceptable for publication, you may indicate that here to bypass the “Comments to the Author” section, enter your conflict of interest statement in the “Confidential to Editor” section, and submit your "Accept" recommendation.

Reviewer #2: (No Response)

Reviewer #3: (No Response)

2. Is the manuscript technically sound, and do the data support the conclusions?

Reviewer #2: Yes

Reviewer #3: Yes

3. Has the statistical analysis been performed appropriately and rigorously? 

Reviewer #2: N/A

Reviewer #3: N/A

4. Have the authors made all data underlying the findings in their manuscript fully available?

Reviewer #2: No

Reviewer #3: No

5. Is the manuscript presented in an intelligible fashion and written in standard English?

Reviewer #2: Yes

Reviewer #3: Yes

6. Review Comments to the Author

Reviewer #2: Thomson et al. conducted semi-structural interviews to 6 treatment-resistant depression (TRD) patients who underwent BNST deep brain stimulation (DBS) and 5 caregivers pre- and post-operatively. The interview topics were related to self-concept, device acceptability, and relationships with caregivers, which were generally overlooked by physicians, and current clinical score (such as HAMD and BDI) do not present these themes. This qualitative study explores in-depth understanding of relationship between alleviation of depression with above experience of DBS for TRD, and provide new insight into a more comprehensive antidepressant effect evaluation of DBS for TRD. As a clinician who is heavily involved in such procedure, I enjoy reading the manuscript, and find it illuminating. Below are some comments I have:

Major:

1. Regarding the candidate who completed pre but not postop interview: this may be a source of bias since the clinical effect may be less satisfactory than those who chose to complete both.

2. No mentioning of clinical efficacy in terms of alleviation in depressive symptoms (HAMD or BDI). I’m curious about change in QoL, too.

3. Line 189-192: How do you define subjectively meaningful vs more modest antidepressant effects?

4. Line 196: Any measurable correlation between decreased depression and transformation in self concept? I understand that the authors are trying to conduct a qualitative study, but as a reader from a neurosurgeon/neuroscientist background, I can’t help but to try to capture some correlations between the two.

5. In the current format, the data is laid out in a relatively mixed fashion. A summary table of patients’ demographic, pre and post depression rating scale score, and personal and relational changes may make it easier for the reader to see the big picture.

6. Line 337: What is the medication/psychotherapy protocol in this trial? Do patients continue to receive best medical treatment during trial period?

Minor

1. Line 53: “microelectrode” is misleading. This is the probe we use in the OR to record spike signals before inserting DBS leads.

2. Line 106: female/male ratio is 5:1. What kind of impact will it have on the study?

3. What is the SD/SEM of age/duration of illness?

4. Line 109: The candidate who chose not to complete postop interview is a patient or caregiver?

5. Line 112: SD/SEM would help.

6. Line 119: Does the location of the interview (home vs research center) or method (in person vs phone vs video conference) have any effect on interview results?

7. Line 246: Which IPG (model type) is used in this study?

8. Line 377: Any measurable correlation between personality restoration and degree of benefit? Is DBS restoring personality that leads to antidepressant effect, or the antidepressant effect brings back the personality, or simultaneously? This is a challenging question, as there is not enough data to support any theory, but would like to know what the authors think.

9. Line 411: Does the time of the relationship between patients and caregivers have any impact on how DBS changes relationships?

10. Recharging is a common cause of patient frustration. As a clinician, I would like to know specifically which part(s) of recharging (length, frequency, method, etc) is most annoying, and how to make it more tolerable (e.g. recharging time reduced from 15 to 5 minutes each day?)

11. Any chance the authors can share the interview recordings and/or the transcript? Of course anonymization is required, but this would help us conduct similar research in the future, too.

Reviewer #3: Thomson and colleagues report on the lived experiences of 6 patients with treatment-resistant depression (TRD) and 5 caregivers before and after DBS. The paper documents the views on self-concept, device acceptability, and relationships before and after DBS. The manuscript I am reviewing seems to be a revision after at least one previous peer review. It might be good to note that I am seeing this manuscript for the first time. This manuscript is a well-written, interesting paper and the used qualitative method gives a broader view on these topics than standardized instruments. Therefore, I think this paper contains new and valuable information, and is a valuable addition to the literature. I have only minor comments that might improve this paper.

1. The results section is filled mostly with words like ‘some’, ‘many’ and ‘others’ without specifying the number of patients or caregivers that named this particular theme. Could the authors specify the number throughout the manuscript?

2. Although the discussion contains an interesting section on personality and identity in treatment-resistant depression, it is very general and missing a link with the presented data in the Results. For instance, what do the data in this paper tell us about and relate to the data found on the NEO-FFI? And what can these data tell us about the improbability of personality restoration? How are the perceived changes in personality similar to or different from those reported by Parkinson or OCD patients?

3. Minor detail: the structure in the results is 1) personality; 2) device acceptability and 3) relationships. In the discussion device acceptability and relationships are discussed in a different order. I think it would be more systematic to adhere to the same order in the discussion as used in the results.

4. I spotted a typo on p.20 (Discussion): ‘Ramasubba’ should be ‘Ramasubbu’.

7. PLOS authors have the option to publish the peer review history of their article (what does this mean?). If published, this will include your full peer review and any attached files.

Reviewer #2: No

Reviewer #3: No

---

## [Author Response · Author response to Decision Letter 1]

26 Jan 2023

Dear editor/reviewers, 

Thank you for taking the time to review our manuscript and for your constructive feedback. Responses to reviewer comments (in bold) and other manuscript amendments are listed below. Page/line numbers align with the revised version of the manuscript containing track changes. Underlined sections reflect new additions to the manuscript. 

Reviewer #2: Thomson et al. conducted semi-structural interviews to 6 treatment-resistant depression (TRD) patients who underwent BNST deep brain stimulation (DBS) and 5 caregivers pre- and post-operatively. The interview topics were related to self-concept, device acceptability, and relationships with caregivers, which were generally overlooked by physicians, and current clinical score (such as HAMD and BDI) do not present these themes. This qualitative study explores in-depth understanding of relationship between alleviation of depression with above experience of DBS for TRD, and provide new insight into a more comprehensive antidepressant effect evaluation of DBS for TRD. As a clinician who is heavily involved in such procedure, I enjoy reading the manuscript, and find it illuminating. Below are some comments I have:

Major:

1. Regarding the candidate who completed pre but not postop interview: this may be a source of bias since the clinical effect may be less satisfactory than those who chose to complete both.

The candidate who did not complete the postop interview had an unsatisfactory clinical response to DBS and this contributed to them not completing the interview (the severity of their depressive illness was extreme). While the absence of their voice in the postop data may bias these findings (although the concept of bias is viewed somewhat differently in qualitative research), the inclusion of their caregiver allowed for some of their postop experiences (which were generally negative) to be incorporated into the findings. 

2. No mentioning of clinical efficacy in terms of alleviation in depressive symptoms (HAMD or BDI). I’m curious about change in QoL, too.

Semi-structured interviews and clinical measures were conducted at a similar timepoint (9-months post-DBS), but were not precisely time locked (conducted on the same day). Given this and patients’ fluctuations in response, we have not commented on this within the paper. Long-term clinical outcome data will be presented in the efficacy study publication (footnote inserted mentioning this). But broadly speaking, the clinical outcomes measures collected at the 9-month timepoint reflected participants’ subjective perceptions. Those who reported little-to-no benefit (n = 3), or slight worsening of depression (n = 1) were non-responders according to MADRS and HAM-D and maintained a ‘severe’ depression status according to BDI. For one patient who reported a subjectively meaningful improvement, this was reflected in both clinician and self-rated measures (MADRS = response (>50% decrease), HAM-D = close to response (45% decrease), BDI = from ‘severe’ to ‘mild’). However, for the other patient who reported a meaningful improvement, self-rated measures reflected their subjective reports (BDI = ‘severe’ to ‘minimal’), but clinician-rated measures did not (MADRS = no response, HAM-D = no response). While the patient experienced distinct improvements in mood and interest, functional recovery was slow and the process of social reintegration challenging. This may account for some of the discrepancy in subjective and objective (clinician-rated) outcomes. Regarding quality of life, according to the Q-LES-Q-SF, QoL scores remained stable for all, with the exception of the one responder whose score increased from 4% to 68%, and a non-responder whose score decreased from 21% to 7%. 

EDIT TO MS [Pg. 5, Footnote] 

 Clinical trial details, including inclusions/exclusion criteria, participant characteristics, surgical information, and full psychometric outcomes, will be presented in a forthcoming publication. Any correspondence regarding the clinical trial should be directed to paul.fitzgerald@anu.edu.au.

3. Line 189-192: How do you define subjectively meaningful vs more modest antidepressant effects?

Clarifying descriptors added to manuscript. 

EDITS TO MS [Pg. 10, Line. 208]

With DBS, patients’ perspectives of themselves were greatly influenced by the procedure’s perceived benefit. Those who experienced a subjectively meaningful antidepressant benefit (i.e., described a pronounced improvement in mood that was sustained over multiple months) conveyed a sense of restored self (n = 2). Those who experienced modest or transient benefits (n = 3) (i.e., described subtle improvements in mood and/or notable lifts lasting only 2-3 days duration) recognised encouraging moments of their prior self (e.g., increased curiosity, interest in work activities).

4. Line 196: Any measurable correlation between decreased depression and transformation in self concept? I understand that the authors are trying to conduct a qualitative study, but as a reader from a neurosurgeon/neuroscientist background, I can’t help but to try to capture some correlations between the two.

My observation based off this small sample is that there was a correlation between decreasing depression and transforming self-concept, with those experiencing more pronounced antidepressant benefit reporting a greater sense of transformation (or restoration) of self. In terms of a measurable correlation, this is outside the aim and scope of this qualitative study. 

5. In the current format, the data is laid out in a relatively mixed fashion. A summary table of patients’ demographic, pre and post depression rating scale score, and personal and relational changes may make it easier for the reader to see the big picture.

A table has been inserted on page 5 that provides an overview of participant demographics. Our reasoning for not including pre and post outcome measures in addressed above (response 2). While we agree it would be helpful to have a table providing full case summaries, we are conscious of protecting participants’ confidentiality. Presenting individual participant characteristics (as opposed to group demographics) alongside qualitative excerpts runs the risk of making individual participants and their comments identifiable (potentially to their spouse, parent, child etc.). We explored alternative methods for laying out the results section; however, the current layout was the best approach for ensuring that the timepoints were distinct (pre/post DBS) and allowing sufficient space for both patient and caregiver experiences to be examined across the three themes. 

6. Line 337: What is the medication/psychotherapy protocol in this trial? Do patients continue to receive best medical treatment during trial period?

The line the above comment is referring to is regarding caregivers’ use of medication/psychotherapy. ‘Caregivers’ has been inserted to provide clarity. 

Edits have been made to the Procedure section regarding patient therapies during the trial. 

EDIT TO MS [Pg. 6, Line. 144]

Any existing treatments (medications/psychotherapy) were kept constant during the blinded phase with participants able to make changes during the open-label follow-up.

Minor

1. Line 53: “microelectrode” is misleading. This is the probe we use in the OR to record spike signals before inserting DBS leads.

Corrected.

2. Line 106: female/male ratio is 5:1. What kind of impact will it have on the study?

Edits have been made to the Limitations section acknowledging that this ratio impacts the transferability of the findings. We are reluctant to make assumptions about how this would have impacted the nature of the findings themselves. 

EDIT TO MS [Pg. 26, Line. 577]

The majority of the patient sample were female (5 of 6), which may reflect the higher representation of females in depressed populations. This gender imbalance is likely to have influenced the nature of the observed findings, potentially reducing the transferability of these males with depression.

3. What is the SD/SEM of age/duration of illness?

This information has been included in Table 1 [Page 5.] 

4. Line 109: The candidate who chose not to complete postop interview is a patient or caregiver?

The candidate was a patient. Edits made to clarify this. 

EDIT MS [Pg. 5, Line. 114]

One DBS candidate who completed a preoperative interview did not complete a postoperative interview as it was deemed too burdensome for the patient.

5. Line 112: SD/SEM would help.

Added to Table 1 [Page 5.]

6. Line 119: Does the location of the interview (home vs research center) or method (in person vs phone vs video conference) have any effect on interview results?

It is possible that the different locations and methods could have affected the interviews and results (e.g., increased anxiety for interviewee in a clinical setting, less rapport developed with interviewer via video conference, absence of visual cues for interviewer over the phone). While it would have been preferable to have consistency in interviewing procedures, logistically this was not possible and our focus was on making participation as simple as possible for participants (recognising that the participation requirements of the concurrent clinical efficacy trial were considerable). 

7. Line 246: Which IPG (model type) is used in this study?

EDIT TO MS [Pg. 6, Line. 139]

Participants then underwent DBS surgery and were implanted with Medtronic Activa 3389 electrodes in the BNST.

8. Line 377: Any measurable correlation between personality restoration and degree of benefit? Is DBS restoring personality that leads to antidepressant effect, or the antidepressant effect brings back the personality, or simultaneously? This is a challenging question, as there is not enough data to support any theory, but would like to know what the authors think.

See above comments to response 4 regarding measurable correlations. On the matter of the directional relationship between depressive symptoms and personality; we suspect that it is the alleviation of the depressive symptoms that allows their personality to emerge, that while depression is severe and sustained, their personality is masked or inhibited. 

9. Line 411: Does the time of the relationship between patients and caregivers have any impact on how DBS changes relationships?

Details about the relationship length have been included in Table 1 [Page 5.]. All had very long-term relationships (range 24 – 50 years). Relationship time did not appear to impact how DBS changed relationships. If the above comment was in fact about relationship ‘type’ rather than ‘time’, some further comments have been included in the manuscript regarding this. 

EDIT TO MS [Pg. 24, Line. 507]

In De Haan and colleagues’ (52) OCD sample, patients reported that both they and their partners had to “get used to” the new them. As OCD had always been present in their relationship, couples had created lives that worked around the illness. For the married couples in the current sample (n = 4), their relationships were established well before the onset of depression. Therefore, partners were experiencing varying degrees of ‘reacquaintance’ with their loved one (depending on benefit gained from DBS). Regardless of outcome, DBS created challenges for marital relationships. These couples, however, had weathered similar storms across their long-term, committed relationships. So too was the case for the parent-child pair. 

10. Recharging is a common cause of patient frustration. As a clinician, I would like to know specifically which part(s) of recharging (length, frequency, method, etc) is most annoying, and how to make it more tolerable (e.g., recharging time reduced from 15 to 5 minutes each day?)

The following comment in the results section speaks to the specific parts of the recharging process that were most annoying for patients: 

“Frustration was associated with difficulty establishing a reliable connection, inconsistency in how long and often recharging was required, recharging taking longer than originally expected, and the growing need to prioritise recharging in daily life.” [Page. 13, Line 207]

Regarding how the process of recharging could be improved or made more tolerable, some suggestions have been added to the discussion (underlined sections below). Please note, specific targets (e.g., reducing time from 2 hours to 1) are not included as this was not explored directly with participants. 

EDIT TO MS [Pg. 23, Line. 485]

The recharging needs for individuals with psychiatric conditions typically differs from those with movement disorders (higher voltages resulting in faster battery depletion), as does their psychological and cognitive profiles. In OCD, post-operative education has been shown to assist with device-related anxiety (55), but further investigation is needed. Notable memory difficulties, which the majority of current patients reported (n = 4), can affect retention of device-related information. For this reason, perioperative device education may require modification for TRD purposes. This may include consideration of when device education is offered (e.g., not immediately after surgery) and conducting multiple sessions. Those who had received support from their device company representative in the months following surgery had found this helpful for improving their recharging process (achieving a more reliable recharging connection). A reduction in how long and how often recharging is required would increase the tolerability of the process. This is dependent on the capacity of the current technology. As DBS for psychiatric indications progresses, so too should device technology and research evaluations of patients’ device-related needs (17).

11. Any chance the authors can share the interview recordings and/or the transcript? Of course anonymization is required, but this would help us conduct similar research in the future, too.

The sharing of interview recordings/complete transcript data (including de-identified) with any person outside of the research team was not a condition participants agreed to when consenting to take part in the study. Participants only agreed to have relevant excerpts of their transcript used as required within the manuscript. Given the particularly sensitive nature of the data (e.g., participants discussing intimate details of their relationship with the patient/caregiver) we are particularly conscious of maintaining participants’ confidentiality. Even after the process of de-identification, it could be possible to establish a participant’s identity from the full transcript data. We do, however, recognise the value of this rich data for other interested researchers and will include this condition in future studies. 

Reviewer #3: Thomson and colleagues report on the lived experiences of 6 patients with treatment-resistant depression (TRD) and 5 caregivers before and after DBS. The paper documents the views on self-concept, device acceptability, and relationships before and after DBS. The manuscript I am reviewing seems to be a revision after at least one previous peer review. It might be good to note that I am seeing this manuscript for the first time. This manuscript is a well-written, interesting paper and the used qualitative method gives a broader view on these topics than standardized instruments. Therefore, I think this paper contains new and valuable information, and is a valuable addition to the literature. I have only minor comments that might improve this paper.

1. The results section is filled mostly with words like ‘some’, ‘many’ and ‘others’ without specifying the number of patients or caregivers that named this particular theme. Could the authors specify the number throughout the manuscript?

We acknowledge that an indicator of frequency as suggested would be helpful; however, we wish to avoid extensive quantification within the results. This is an argument made by many qualitative researchers, in part because each participant interview is unique, and the questions asked within them (beyond the interview guide) can differ. Quantification suggests only a certain number had a particular experience, when in fact others may have also. The nature of that interview and the questions asked may not have led to it being discussed. We provide an editorial reference on this issue from Professor Joanne Neale and colleagues in the journal Addiction (Prof Neale is the Associate Editor for Qualitative Research at Addiction). 

Neale, J., Miller, P., & West, R. (2014). Reporting quantitative information in qualitative research: Guidance for authors and reviewers. Addiction, 109, 175-176. 

Some semi-quantification with non-specific terms is used e.g., “a few”, “some”, “many” to indicate frequency in a relative sense. Single cases are remarked upon but are typically presented in a way that does not exclude others from potentially having had the same experience. We have provided explicit frequencies when reporting types of personality change, our reason being that each participant was asked the same specific questions on this topic to prompt these responses. We have also provided explicit frequencies for participants’ perceived benefit from DBS, as we have found this to be a particular point of interest for readers and reviewers in the past, who request these frequencies specifically. 

2. Although the discussion contains an interesting section on personality and identity in treatment-resistant depression, it is very general and missing a link with the presented data in the Results. For instance, what do the data in this paper tell us about and relate to the data found on the NEO-FFI? And what can these data tell us about the improbability of personality restoration? How are the perceived changes in personality similar to or different from those reported by Parkinson or OCD patients?

We agree that parts of this section are too general and require linking back to the study findings. This particular sub-section (Personality and Identity in Treatment Resistant Depression) has been re-ordered and edited in order to make these link between the broader literature and the current findings clearer. 

3. Minor detail: the structure in the results is 1) personality; 2) device acceptability and 3) relationships. In the discussion device acceptability and relationships are discussed in a different order. I think it would be more systematic to adhere to the same order in the discussion as used in the results.

As suggested, the discussion sections have been re-ordered to align with the themes in the results.

4. I spotted a typo on p.20 (Discussion): ‘Ramasubba’ should be ‘Ramasubbu’.

Corrected.

---

## [Decision Letter · Decision Letter 2]

15 Mar 2023

PONE-D-21-34707R2Personal and relational changes following deep brain stimulation for treatment-resistant depression: A prospective qualitative study with patients and caregiversPLOS ONE

Dear Dr. Thomson,

Thank you for submitting your manuscript to PLOS ONE. After careful consideration, we feel that it has merit but does not fully meet PLOS ONE’s publication criteria as it currently stands. Therefore, we invite you to submit a revised version of the manuscript that addresses the points raised during the review process. The presence of quantitative results is little bit confusing to the reader. It seems that the study is a mixed-method not a qualitative one. I do suggest describe the participants' characteristics without offering quantitative data.   

We look forward to receiving your revised manuscript.

Kind regards,

Fatma Refaat Ahmed, Ph.D.

Academic Editor

PLOS ONE

Journal Requirements:

Reviewers' comments:

Reviewer's Responses to Questions

**Comments to the Author**

1. If the authors have adequately addressed your comments raised in a previous round of review and you feel that this manuscript is now acceptable for publication, you may indicate that here to bypass the “Comments to the Author” section, enter your conflict of interest statement in the “Confidential to Editor” section, and submit your "Accept" recommendation.

Reviewer #3: All comments have been addressed

Reviewer #4: All comments have been addressed

2. Is the manuscript technically sound, and do the data support the conclusions?

Reviewer #3: Yes

Reviewer #4: Yes

3. Has the statistical analysis been performed appropriately and rigorously? 

Reviewer #3: N/A

Reviewer #4: Yes

4. Have the authors made all data underlying the findings in their manuscript fully available?

Reviewer #3: No

Reviewer #4: No

5. Is the manuscript presented in an intelligible fashion and written in standard English?

Reviewer #3: Yes

Reviewer #4: Yes

6. Review Comments to the Author

Reviewer #3: All my posted comments have been addressed by the authors. Thank you for this very interesting paper.

Reviewer #4: This is the first time I have seen this manuscript but it appears to be the second revision. I read the previous reviewer comments, which I think were very reasonable. I think the authors have responded well to the points raised. Overall, this is an important study in a field that needs more data. The methodological background to the analysis is well described and I felt the findings were easy to read and insightful with illustrative quotes. I enjoyed the discussion. Please publish this paper with no further revisions.

7. PLOS authors have the option to publish the peer review history of their article (what does this mean?). If published, this will include your full peer review and any attached files.

Reviewer #3: No

Reviewer #4: No

---

## [Author Response · Author response to Decision Letter 2]

17 Mar 2023

Dear Academic Editor

Thank you for reviewing our revised submission. Below is our response to your recommendation for our manuscript. 

The presence of quantitative results is little bit confusing to the reader. It seems that the study is a mixed-method not a qualitative one. I do suggest describe the participants' characteristics without offering quantitative data. 

Additional participant information in the form of a table containing quantitative participant demographic data was introduced following recommendations from a reviewer. Prior to this, we had described participant characteristics within text, using minimal quantitative data. Given your above recommendation, we have decided to revert to our previous approach. Table 1 has been removed, with details provided within the ‘Participants’ text.

---

## [Editor Report · Decision Letter 3]

27 Mar 2023

Personal and relational changes following deep brain stimulation for treatment-resistant depression: A prospective qualitative study with patients and caregivers

PONE-D-21-34707R3

Dear Dr. Thomson,

We’re pleased to inform you that your manuscript has been judged scientifically suitable for publication and will be formally accepted for publication once it meets all outstanding technical requirements.

Kind regards,

Fatma Refaat Ahmed, Ph.D.

Academic Editor

PLOS ONE
---

## [Editor Report · Acceptance letter]

29 Mar 2023

PONE-D-21-34707R3 

Personal and relational changes following deep brain stimulation for treatment-resistant depression: A prospective qualitative study with patients and caregivers 

Dear Dr. Thomson:

I'm pleased to inform you that your manuscript has been deemed suitable for publication in PLOS ONE. Congratulations! Your manuscript is now with our production department. 

Kind regards, 

on behalf of

Dr. Fatma Refaat Ahmed 

Academic Editor

PLOS ONE